# Appropriate Technologies to Accompany Sunscreens in the Battle Against Ultraviolet, Superoxide, and Singlet Oxygen

**DOI:** 10.3390/antiox9111091

**Published:** 2020-11-06

**Authors:** Paolo U. Giacomoni

**Affiliations:** Insight Analysis Consulting, Madison, AL 35758, USA; paologiac@gmail.com

**Keywords:** ultraviolet, superoxide, singlet oxygen, cell blebbing, skin aging, peroxidative cascade, antioxidants

## Abstract

The interaction of ultraviolet radiation with biological matter results in direct damage such as pyrimidine dimers in DNA. It also results in indirect damage provoked by the production of reactive oxygen species (ROS) catalyzed by photosensitizers. Photosensitizers can be endogenous (e.g., tryptophan) or exogenous (e.g., TiO_2_ and other photostable UVA sunscreens). Direct damage triggers an inflammatory response and the oxidative and proteolytic bursts that characterize its onset. The inflammatory reaction multiplies the effects of one single photon. Indirect damage, such as the peroxidative cascade in membrane lipids, can extend to thousands of molecular modifications per absorbed photon. Sunscreens should therefore be formulated in the presence of appropriate antioxidants. Superoxide and singlet oxygen are the main ROS that need to be tackled: this review describes some of the molecular, biochemical, cellular, and clinical consequences of exposure to UV radiation as well as some results associated with scavengers and quenchers of superoxide and singlet oxygen, as well as with inhibitors of singlet oxygen production.

## 1. Introduction

About half of the solar energy reaching the earth is made of visible photons. Solar radiation is essential to life on our planet. The chlorophyllean photosynthesis allows organisms of the vegetal kingdom to generate energy by harvesting photons in the visible range, and chromophores within living cells absorb visible light, convert it into thermal energy, and help maintain the temperature within limits that favor the development of the forms of life we know so well.

About 5% of the solar energy reaching the earth is made of “ultraviolet” photons with wavelengths comprised between 290 and 400 nm. These wavelengths are labeled as UVA-1 (400–340 nm), UVA-2 (340–315 nm), UVB (315–290 nm). Shorter ultraviolet wavelengths (UVC) are filtered off by the ozone layer.

It is worth noticing that, with a variability dictated by the hour of the day, the day of the month, the altitude, and the presence of clouds, UVA constitutes about 95% of UV, and UVB constitutes about 5% of UV. It is also worthwhile noticing that UVB, originally defined as the range of wavelengths absorbed by Mylar glasses, is also the range of ultraviolet radiation that is absorbed by pure DNA. This makes it particularly interesting for the potential effects on living organisms, in particular on human skin.

The outcomes of exposure of skin to ultraviolet radiation have been studied for over a century, and the results have been reported in specialized publications and monographs. Among the major biochemical and clinical effects of UV radiation, one finds DNA damage [1,2], protein oxidation [3], lipid peroxidation [4,5], erythema [6,7,8,9,10], pigmentation [11,12], aging [13,14], cancer [15,16], and immune depression [17,18].

The mechanisms of UV damage have been elucidated. DNA (about 0.1% of cellular material) and RNA (about 1% of cellular material) are the main targets of UV-B: one can grossly estimate that the molar extinction coefficient of nucleotides at 290 nm is about 1000, whereas the molar extinction coefficient of amino-acids (about 10% of cellular material) at 290 nm is about 50. Lipids do not have chromophores in the UV-B. Pure DNA, RNA, proteins, or lipids do not absorb in the UVA whereas the complexes formed by DNA with transition metals such as iron do absorb slightly in the UVA. This accounts for the nicking of DNA by UVA in the presence of iron and oxygen [19].

The observation that solar UVB radiation does not penetrate in the dermis is in keeping with the high concentration of nucleic acids and proteins in the epidermis. UVA radiation does penetrate up to a point. Wavelengths not absorbed by biological material, such as 700–650 nm (i.e., red light), can penetrate in the skin and cross the thickness of a hand, as it can be shown with a flashlight in the dark.

Direct absorption of UVB by proteins does not induce stereochemical modifications. Protein modifications are mainly the consequences of photosensitized oxidation [3]. On the other hand, when DNA or RNA absorbs UVB, several base modifications can occur. In particular adjacent pyrimidines can photoreact and form photoproducts, the most frequent being cyclo-butane- and 6-4 Py-Po-pyrimidine dimers (called in short, CBD and 6-4, respectively). CBD and 6-4 provoke stereochemical changes in the double stranded nucleic acid. The removal of pyrimidine dimers and of other DNA damage requires the removal of a short DNA segment that contains the damage, and the re-synthesis of the short sequence. This process is not error-free, and it is generally understood to be perhaps the major cause of the mutagenic and carcinogenic properties of ultraviolet radiation.

Pyrimidine dimers in DNA have been shown to trigger an inflammatory reaction [20]. Such a reaction has been documented by histological analysis to occur in human skin after UV exposure [21]. The inflammatory reaction is accompanied by the proteolysis of the elastic fibers performed by the matrix metallo-proteinases (MMP) that are secreted by fibroblasts upon release of cytokines by the immune cells while they are chemotactically driven across the dermis to reach the UV-damaged cells [22]. The inflammatory response is also accompanied by the oxidative bursts of hydrogen peroxide released by immune cells when crossing the walls of the blood vessels to enter the dermis and when digesting the UV-damaged cell before engulfing its debris. There is also a release of singlet Oxygen when the immune cells cross the dermis [23].

In addition to this mechanism of release of protein-damaging proteases and of pro-oxidants, photosensitization accounts for the production of several other reactive oxygen species. Tyrosine and tryptophan exposed to UV generate hydrogen peroxide [24] that in the presence of transition metals is converted in the hydroxyl radical. In addition, several photosensitizers transfer charge or energy to molecular Oxygen to form Superoxide and singlet Oxygen [25,26]. All of these ROS can have devastating effects on cell membranes because they can trigger the peroxidative cascade in lipids [4,5]. It is worth noticing that the photon absorbed by a photosensitizer and producing one molecule of reactive Oxygen species does indeed provoke thousands of molecular reactions by catalyzing the peroxidative cascade. The peroxidative cascade triggered by a photosensitizer dramatically multiplies the damage, that would have been of one pyrimidine dimer if the photon itself had been absorbed by DNA. Of course, in this case, the inflammatory reaction would have multiplied the damage because of macrophages and other immune cells releasing reactive Oxygen species and matrix metallo-proteinases. This is to say that the absorption by a photosensitizer or by a DNA molecule ends up in generating multiple damage caused by the peroxidative cascade of the inflammatory reaction.

It thus appears that to limit the damage provoked by the exposure of the skin to ultraviolet radiation, one should set up: 1- a technology to reduce the number of UV photons actually impinging on the skin, thus reducing the rate of formation of direct-absorption induced damage (such as pyrimidine dimers) as well as 2- a technology to limit the secondary effects of radiation, that is, appropriate antioxidants and scavengers of ROS, or inhibitors of their formation. This is particularly important because singlet Oxygen is formed not only by endogenous photosensitizers, but also by commercial sunscreens such as titanium dioxide and zinc oxide [27] and terephthalylidene dicamphor sulfonic acid (a sunscreen whose trade name is Mexoryl SX) [28].

This papers deals with the molecular, biochemical, cellular, and clinical effects of superoxide scavengers such as α-tocopherol (that is oxidized to produce α-tocopheryl quinone, α-tocopherol dimer, and α-tocopherol dihydroxy dimer), singlet Oxygen scavengers such as Xanthine and β-carotene, singlet oxygen quenchers such as 3-(4-hydroxy-3-methoxybenzyl)pentane-2,4 dione (Acetyl Zingerone), phenol-4-[(1E,3S)-3-ethenyl-3,7-dimethyl-1,6-octadienyl] (Bakuchiol), as well as of Bis (Cyano Butylacetate) Anthracenediylidene, (Micah), an inhibitor of the formation of singlet Oxygen.

The reactive oxygen species relevant to skin damage are summarized in Table 1.

## 2. Cellular Consequences of UV-Radiation

The chemistry of the peroxidative cascade of lipids will be described in more detail below. It is important here to evoke a frame of thinking to grasp, albeit in an approximate way, why lipid peroxidation is of paramount importance in cell physiology. As a simplified statement, one could say that in living organisms, the vast majority of lipids are confined to cell membranes. Membranes can be visualized as impermeable walls formed by phospholipid bilayers that keep the inside “in” and the outside “out”, thus allowing water-soluble molecules within the cell to perform all of their tasks without the risk of being diluted out in the aqueous environment. The phospholipids in a membrane are so oriented that the charged phosphate heads are in contact with water molecules in the cytoplasm and in the external environment, while the lipid tails associate in hydrophobic bonds at the interior of the cellular wall. The access of foreign molecules to the cytoplasm is regulated either by appropriate “pumps” or by the complex mechanisms of endocytosis or pinocytosis. The similarly complex mechanisms of exocytosis allow the secretion of molecules from the interior of the cells towards the environment. Rare are the molecules that can cross the membrane freely: one of them is the protonated superoxide radical that does not carry a charge. With very few exceptions, to enter a cell, charge-carrying molecules that cannot use an existing “pump”, have to be entrapped in lipid vesicles called liposomes, thus mimicking the process of endocytosis.

When a cell membrane interacts with a reactive oxygen species, the molecules in the lipid bilayer are the preferred targets. In the process of the peroxidative cascade, they bind molecular oxygen and acquire a defined electric polarity. The energy potential minimum of a system containing peroxidated lipid tails will no longer be observed in a structure containing hydrophobic bonds. Peroxidated molecules will find a potential minimum in water, and to reach that minimum peroxidated molecules will move towards water and the hydrophobic structure will be disrupted, thus creating havoc in the cellular membrane.

A most remarkable phenomenon that can be observed in cultured cells exposed to UV is the onset of zeiosi, also called blebbing [29]. Blebbing is a morphological phenomenon provoked by oxidative stress [30] that can be analyzed by microscopical and ultrastructural technologies. Blebbing consists of the appearance of bubbles on the surface of the cells, hence the name zeiosis (in Greek: boiling) reminiscent of the bubbles generated in boiling water. It is a cytoskeleton-linked phenomenon that has been shown to occur in human epidermis in vivo [31]. The presence of blebs in vivo, in UV irradiated epidermis, was pointed out with an accurate analysis of transmission electron microscopy. That analysis was aimed at finding evidence of blebs linked to the main cellular body. Often in transmission electron microscopy experiments, such evidence is difficult to find because the microtome does not always have the chance of cutting through the bridge connecting blebs and cellular body.

Zeiosis is the result of a comprehensive phenomenon of cellular response to stress and damage [32]. Histologically it gives rise to the phenomenon of spongiosis, the appearance of isolated granules in the interstitial space that is generated after the rounding up of the cells as a consequence of UV stress. [31]. It has been suggested that the “granules” in the interstitial space are indeed parts of blebs, cut by the microtome “off” the bridge linking the bleb to the main body of the cell. Since it occurs both in culture and in vivo, it is reasonable to use the blebbing as an easy-to-observe endpoint to determine the efficacy of stressing or of antistress treatments. Blebbing after UV is a cytoskeleton-dependent phenomenon, observed in A 431 epidermoid cells as well as in cultured normal human keratinocytes [29,33]. The normal morphology of human cultured keratinocytes that is, polygonal cells with short microvilli, is dramatically altered by UV radiation. The actin filaments are rearranged and lose their organized structure [33]. The same, albeit to a lesser extent, occurs to tubulin filaments [32]. These major cytoskeletal alterations are in keeping with the detachment of the cells from the substrate that occurs hours after the exposure to ultraviolet radiation.

Another consequence of UV exposure in cultured cells is the fragmentation of chromatin, that is considered to be a specific indicator of cellular apoptosis [32] and this allows one to quantify overall cellular damage by counting cells that have undergone blebbing, and to quantify apoptosis by counting cells with fragmented chromatin.

## 3. Antioxidants and Ultraviolet Radiation

The reactive oxygen species produced by the interaction of ultraviolet radiation with biological matter together with some appropriate antioxidants are reported in Table 2.

Let us recall that one process of ROS generation is the consequence of the photosenzitizer-induced charge transfer that produces the superoxide anion [5]. In this process many steps are necessary to generate one lipid peroxide. Indeed O_2_^•–^ can reduce ferric iron or cupric copper to their reduced states. Via the Fenton reaction, reduced transition metals split H_2_O_2_ into OH- and OH* and the hydroxyl free radical can strip a hydrogen atom from a lipid molecule to generate water and a lipid free radical L-H + OH* >> L* + H_2_O. In the presence of molecular oxygen, a lipid peroxide radical is formed L* + O_2_ >> L-O-O* that can strip protons from nearby lipids thus propagating the peroxidative cascade.

The production of the peroxy radical is, therefore, the result of the action of the hydroxyl radical. Since the hydroxyl radical is the result of the Fenton reaction where superoxide reduces a transition metal that in turn converts hydrogen peroxide into hydroxyl radical and hydroxyl anion [5], we will consider that vitamin E (α-tocopherol) is an “antidote” against UV-generated superoxide, hydroxyl radical, and peroxy radical.

Another reactive oxygen species generated via photosensitization is singlet Oxygen. The generation of singlet Oxygen is the consequence of energy transfer from an excited photosensitizer to molecular Oxygen. Singlet Oxygen, too, induces peroxy radicals. Preferred targets of singlet Oxygen are the carbon atoms linked by a double bond in alkenes. That is to say that, at variance with the chemical process following the generation of superoxide by charge transfer discussed above, in the presence of photosensitizer-induced singlet Oxygen, an unsaturated lipid molecule turns into a lipid peroxide L-O-O-H, in one single step and without the intervention of free radicals [5].

L-H + ^1^O_2_ >> LOOH

In the presence of a reduced transition metal, the lipid peroxide will produce a lipid oxide and the hydroxyl anion while oxidizing the transition metal

L-O-O-H + Fe2 >> L-O* + OH- + Fe3

L-O* can strip a Hydrogen atom from a nearby lipid molecule generating a lipid free radical.

L-O* + L-H >> L-O-H + L*

In the presence of molecular Oxygen, the lipid free radical will generate a lipid peroxide able to strip a Hydrogen atom from a nearby lipid and trigger the peroxidative cascade.

L* + O_2_ >> L-O-O*

Alternatively, L-O* can react with molecular Oxygen to give O-L-O-O*, a molecule able to strip a Hydrogen atom from a nearby lipid molecule, resulting in one lipid peroxide and one lipid radical O-L-O-O* + L-H >> O-L-O-O-H + L* and the lipid radical L* can in turn react with molecular Oxygen to generate L-O-O* that can strip a Hydrogen atom from nearby lipids etc.

Lipid peroxides can be scavenged by α-tocopherol, Acetyl Zingerone, or Bakuchiol. ^1^O_2_ can also be scavenged by carotenoids or tryptophan. Its production can be hindered by appropriate molecules, such as Micah and Acetyl Zingerone, that will also be discussed. It has to be noted that scavenging singlet Oxygen might not be sufficient. It has been pointed out that the presence of α-tocopherol made things worse when a ^1^O_2_-generating sunscreen was tested. One can indeed surmise that singlet Oxygen could be reacting with α-tocopherol and trigger the singlet Oxygen-induced peroxidative cascade described above.

Another ROS, peroxynitrite, is an oxidant produced by the reaction between the endogenously produced nitric oxide NO and the photosensitizer-produced superoxide radicals O_2_^•–^ that is effective in inducing lipid peroxidation, damaging DNA, and oxidizing sulfhydryl groups in proteins. As such, peroxynitrites could be damaging for the cytoskeleton, one of the indirect targets of the nefarious action of ultraviolet radiation on cells [29,30,33]. Often neglected by the skin care industry, nitric oxide is a vaso-dilating agent generated by inducible and constitutive nitric oxide synthase within cells, often involved in unwanted phenomena such as cutaneous itch. Inducible NO synthase produces NO as a defense mechanism in response to cytokines such as IL-1 or TNF-α that are secreted upon infection or other aggressions. It might be of interest to learn that UVB-exposure induces the expression of the iNOS in vessel endothelia of normal human skin and in cultured human dermal endothelial cells and that exposure to ultraviolet A1 (50 J/cm^2^) in the absence of cytokines led to the expression of nitric oxide synthase-2 in human skin organ cultures.

It has been reported that UVA as well as UVB do induce inducible endothelial NO synthase following two different mechanisms. Irrespective of the mode of production of nitric oxide, though, peroxy-nitrite formation is dependent on the presence of the superoxide anion. Therefore, in this review we will confine ourselves to scavengers or quenchers of superoxide, keeping in mind that the scavenging of peroxinitrite will be a windfall profit for scavengers of superoxide.

## 4. UV and α-Tocopherol

Cultured A-431 epidermoid cells were exposed to 1200 J/m^2^ UVB in the presence or the absence of α-tocopherol. In the presence of α-tocopherol, nonirradiated cells did not differ from the untreated control. On the other hand, 24 h after irradiation, many cells detach from the substrate and the ones remaining attached undergo rounding. Of these, 80% display blebbing. When cells are treated with α-tocopherol before, during, and after the exposure to UVB, there is a pronounced inhibition of cell detachment (a marker of cell death) and of cell retraction and rounding. Similar results are observed when α-tocopherol is added only after the irradiation, irrespective of the time of addition, up to at least 7 h after irradiation [34]. These results seem to indicate that the kinetics of formation of the oxidative damage leading to morphological modifications and cell death is relatively slow, insofar as it can be quenched by adding α-tocopherol even 7 h after irradiation.

Cultured normal human keratinocytes were exposed to either 1200 J/m^2^ UVB or 200,000 J/m^2^ UVA (with a filter system that reduced to practically nil the contamination by UVC).

24 h after UVA irradiation, 75% of the cells were damaged (blebbing) and 60% of the cells were apoptotic (chromatin fragmentation).

24 h after UVB irradiation, 75% of the cells were damaged and 95% were apoptotic. The addition of α-tocopherol either before or before and after UVA irradiation totally abolishes cell blebbing but only partially reduces apoptosis if at all. The addition of α-tocopherol either before or before and after UVB irradiation reduces cell blebbing by about 60% and apoptosis by about 30% [33].

The epidermis of 12 human volunteers was either untreated or treated with a topical formulation containing α-tocopherol acetate in nanocapsules and exposed to two individual MED from a solar simulator. Biopsies were taken before and 2, 4, and 7 h after exposure to UV. The analysis of biopsies allowed the morphological changes induced by UV to be detected. Modifications of intercellular interactions and changes in nuclear morphology were apparent 2 and 4 h after irradiation. By 7 h after exposure, the epithelium seemed to have partially recovered its structure and morphology. Treating the epidermis before the exposure to UV with α-tocopherol acetate in nanocapsules abolished the early morphological modifications induced by UV [31]. Pictures of the histology sections from this study are published in [35].

## 5. Singlet Oxygen

Of all the known reactive oxygen species, singlet Oxygen is a very reactive moiety. It can be surmised that singlet Oxygen plays a major role in the generation of UV-induced damage and in accelerating the process of accumulation of damage that is, accelerating the rate of skin aging.

One of the major clinical signs of skin aging is dysfunctional dermis (cutis rhomboidalis, sagging of the skin, elastosis, Milian skin type, Favre Racouchot syndrome, etc.). Dysfunctional dermis in UV-exposed cutaneous tissues has been explored in detail.

One of the ultrastructural signs of photoaging is the disorganization of the elastic fibers [13,14]. This is thought to be the consequence of proteolysis in the extracellular matrix followed by a slow remodeling due to the long time of half renewal of collagen1. Experiments performed with UVA-irradiated cultured human fibroblasts as well as with UVA-irradiated human epidermis show that the exposure to UVA induces the expression of collagenase, a protease responsible for the digestion of dermal elastic fibers [36], Similar results were obtained when cultured cells were exposed to singlet Oxygen [37]. It was therefore reasonable to surmise that singlet Oxygen is the mediator between UVA irradiation and the digestion of elastic fibers.

Cultured human fibroblasts were exposed to singlet Oxygen, generated by thermodissociation of the endoperoxide of the disodium salt of 3,3′-(1,4-naphthylidene) dipropionate (NDPO2). This induced the accumulation of collagenase mRNA in a dose dependent manner. The increase in collagenase expression after singlet oxygen exposure generated with 3 mM NDPO2 was equivalent to that observed with 200–300 kJ/m^2^ UVA and followed similar kinetics. Further evidence for the role of singlet oxygen in the UVA induction of collagenase comes from studies using singlet oxygen enhancers or quenchers. Deuterium oxide increases singlet oxygen lifetime. Incubation in D2O led to an additional increase in steady-state levels of collagenase mRNA after exposure to NDPO2 or to UVA irradiation. In contrast, incubation in sodium azide, a potent quencher of singlet Oxygen, almost totally abrogated the induction of collagenase after exposure of fibroblasts to NDPO2 or to UVA irradiation [38].

It was therefore of interest to ascertain whether well known singlet Oxygen scavengers such as β-Carotene [5] could indeed abrogate the effects of singlet Oxygen. Experiments were performed with cultured human HaCat keratinocytes pretreated with 1.5 µM β-Carotene and exposed to UVA. It was observed that of the 568 UVA-regulated genes, β-Carotene reduced the UVA effect for 143, enhanced it for 180, and did not interact with 245 UVA-regulated genes. In irradiated cells, expression profiles indicate that the presence of β-Carotene inhibits the degradation of the extracellular matrix induced by UVA [39].

In addition to inducing interstitial collagenases, matrix metallo-proteinases and other proteases, singlet Oxygen plays a role in the UV-induced oxidative damage on lipids and proteins [40], and can therefore provoke membrane damage and play a role in the onset of necrosis after UV. Xanthine, a singlet Oxygen scavenger [41], was shown to be effective in avoiding the formation of sunburn cells in mice skin explants when applied at concentrations between 1 and 10 mM during irradiation with 600–1200 J/m^2^ UVB [42].

Other singlet Oxygen scavengers or quenchers are worthwhile being quoted. Bakuchiol has been shown to be an effective scavenger of singlet Oxygen generated using hydrogen peroxide and the Molybdate anion. As such it can be surmised that it could inhibit the singlet Oxygen-induced peroxidation of squalene, thus hindering the clogging of the pores and counteracting the onset of acne. In a pilot study with 54 volunteers in four groups (A: control, B: 2% salicylic acid, C: 1% Bakuchiol and D: 2% salicylic acid and 1% Bakuchiol) it was shown that Bakuchiol alone clears 42% of the acne in 4 weeks, whereas salicylic acid clears 34% and the combination of the two clears 48% of the acne. In the untreated control group, the clearing of acne was 5% after 4 weeks [43]. Bakuchiol has also been tested for 12 weeks on 50 healthy volunteers aged 40–55 as an anti-photoaging agent when formulated at 0.5% in topical creams. It appeared that Bakuchiol significantly decreases the surface of the area covered with wrinkles, and that it reduces hyperpigmentation. The experiment was performed against 0.5% retinol, and no difference in the efficacy of the two actives was observed [44].

Another quencher of singlet Oxygen is Acetyl Zingerone [45]. This molecule has been shown to (i) significantly attenuate intracellular ROS in cultured keratinocytes under UVA radiation [45], (ii) inhibit the post-UV formation of pyrimidine dimers in melanocytes (a process that continues taking place after the end of UV irradiation) possibly by scavenging peroxynitrite as a key precursor in the process [45,46], (iii) improve extracellular matrix integrity by repressing matrix metalloproteases [47], and (iv) visibly improve the signs of facial photoaging (wrinkle, uneven pigmentation, and redness) following daily application of lotion containing 1% AZ over 8 weeks compared with its placebo lotion [48].

The use of singlet Oxygen scavengers such as carotenoids has at least two drawbacks, a physical-chemical one and an aesthetic one. In spite of the very high rate of reaction of the scavenger, some singlet oxygen molecule can escape the scavenger and trigger the peroxidative cascade. In addition to this, the intense yellow-orange color of carotenoids might hinder their use in topical cosmetic products.

A winning strategy could be to hinder the formation of singlet Oxygen instead of trying to scavenge it out of the skin exposed to UV. Porphyrins are ubiquitous in human skin [49], and are well-known generators of singlet Oxygen upon exposure to solar radiation. Aromatic cyano-acrylates were tested for their capability to quench the singlet and triplet excited states of Protoporphyrin IX. One of the fused-ring Cyano-Acrylates used in the study appeared to quench the triplet state of Protoporphyrin IX exposed to visible radiation, and to hinder the production of singlet oxygen [50].

This molecule, Bis (Cyano Butylacetate) Anthracenediylidene (trade name: Micah), has been tested first on reconstructed epidermis, then on female skin explant, and then in vivo on the skin of 10 volunteers.

Reconstructed epidermis from SkinEthic (Lyon, France) was treated with Micah for 60 min and then exposed to 20 J/cm^2^ of UV from a solar simulator. Samples were fixed in formalin, either immediately, 5 or 10 h after irradiation. After embedding in paraffin, the samples were sliced with a microtome, paraffin was extracted from the histology sections at 65 °C for 30 min followed by repeated extractions with xylol and rehydrated. After 3 min incubation with NaOH, the sections were prepared for the detection of 8-OH-Guanine with the appropriate primary and secondary antibodies and analyzed for the red staining at the site of the target antigen recognized by the primary antibody. The intensity of the red color was larger in the irradiated control than in the nonirradiated control, and the intensity of the Micah-treated irradiated samples was identical to the intensity of the nonirradiated control at 0, 5, and 10 h after UV irradiation.

Female skin-explants were treated with Micah at different concentrations (0.02%–0.2%) in a formula that was applied at 2 mg/cm^2^ for 30 min and exposed to 10 J/cm^2^ UVA. Positive controls were 5% Avobenzone or 2% vitamin E. The amount of free radicals detected after the irradiation of untreated or vehicle-treated samples was more than twice (218% and 239%, respectively) the amount detected in the unirradiated controls. In the samples treated with 0.2% Micah the amount of the free radicals produced was 70% of the unirradiated control, that is, 30% less than the negative control. The samples treated with Avobenzone or vitamin E produced about 60% and 40% more free radicals, respectively, than the ones detected in the unirradiated samples.

The level of IL-1A found in the Micah-treated (0.1% and 0.2%) irradiated samples was practically identical to the one in the nonirradiated controls and about 50% of the level detected in the irradiated untreated controls. Vitamin E and Avobenzone afforded similar results. The same was observed for the level of IL-6. 8-OH-dG increased more than twofold after irradiation in the non-treated samples, and was found to be equal to the level in nonirradiated samples for the Micah-treated (0.2%) exposed to UVA. It is worthwhile noticing that in samples treated with Micah (0.1% and 0.04%) the level is larger than in the nonirradiated control but is smaller than in the untreated, irradiated control, and very similar to the levels found in the irradiated samples treated with Vitamin E or Avobenzone. Last but not least, the level of matrix metallo-proteinase (MMP-1) found in Micah-treated, irradiated samples is slightly smaller than in the nonirradiated control and 30% less than in the UVA exposed control.

Before extending the analysis on human volunteers, the safety of Micah was assessed by the usual tests: genotoxicity (Ames test), ocular irritation (HET-CAM), phototoxicity (MTT ex vivo and in vivo on 22 volunteers), and sensitization (human repeated insult patch test on 50 volunteers).

For the in vivo test, four zones of the skin of 10 volunteers were used: untreated control; UV-irradiated control; non-UV-irradiated with Micah; UV-irradiated with Micah. Micah was applied twice a day for 2 weeks and then the skin was exposed to ultraviolet radiation. Immediately and 24 h after the exposure to 20 J/cm^2^ UVA, suction blisters were performed. The interstitial fluid was analyzed for IL-6 and Matrix Metallo Proteinase 1, the products of two genes that are upregulated by singlet Oxygen. The roof of the blister was analyzed for 8-OH deoxy Guanine, an oxidative product of DNA, catalyzed by singlet oxygen. It was observed that when the UV exposure was performed in the Micah-treated zones, the levels of IL-6 and MMP 1 were remarkably reduced as compared to the controls irradiated in the absence of Micah. In the zone exposed to UVA in the presence of Micah, the level of 8-oxo Guanine was zero. (Hallstar information via personal communication. I had the possibility of reading and discussing the reports from the laboratory that performed the analyses).

## 6. Conclusions

Ultraviolet radiation was known to produce cytoskeletal damage [51,52,53]. It was also known that chemically induced oxidative stress to cytoskeletal protein provoked surface morphological changes and surface blebbing [30,54]. It was therefore to be expected, that UV radiation could induce cell blebbing in cultured cells as well as in vivo, and that was found to be actually the case [29,31].

It was also known that ultraviolet radiation provokes the degradation of NAD in cultured cells as well as in mammalian epidermis [55,56] with the consequent impairment of ATP synthesis that was later observed also in UV-irradiated human skin [57] with an expected drop of membrane potential.

All these phenomena are strongly indicative that cell membranes are the ultimate target of a plethora of actions that can combine in provoking cell death, necrosis, and inflammation. The mechanisms committed to this process are being understood in depth.

We understand now that one single ultraviolet photon absorbed by cellular DNA can provoke one single DNA damage, for instance a pyrimidine dimer. Damaged DNA is generally repaired by the DNA repair mechanism within the cell. Once in roughly one million repair actions, the repair of DNA is not successful, and a mutation is produced. In a long term process, should this mutation be in a specific gene, and should the same cellular DNA accumulate mutations in other control genes, the sum of these mutation can lead to cell transformation and cancer. We also know that one pyrimidine dimer, before being removed, can trigger the onset of an inflammatory reaction, that releases fiber digesting proteases as well as reactive oxidative species. We therefore understand how one single damage (a pyrimidine dimer) can quickly provoke a large number of damaged molecules resulting in protein carbonylation, proteolysis of elastic fibers, damage to neighboring cells, and so forth.

We also understand that upon absorption of an UV photon by a photosensitizer, an electron or an energy quantum can be transferred to molecular oxygen thus generating superoxide ions or singlet oxygen molecules, that can trigger the lipid peroxidation cascade, generate thousands of damaged molecules and create havoc in cellular membranes, thus provoking necrosis and further the stimulation of the inflammatory process.

It is therefore self-evident that sunscreens should be accompanied by scavengers of superoxide ions as well as by scavengers or quenchers of, as well as inhibitors of the production of, singlet oxygen. The personal care industry was quick in understanding the role of free radicals in general, in the process of ultraviolet damage, inflammation, and skin aging. Surprisingly, though, the role of singlet oxygen has been overlooked and technologies appropriate to tackle the cascade of damage triggered by it have only recently been developed. A couple of very interesting technologies to control singlet Oxygen are now at hand and they could be advantageously used in combination with sunscreens, vitamins, and other antioxidants in antiaging formulations.

The FDA seems inclined to allow as sunscreens only TiO_2_ and ZnO, that have been shown to catalyze the generation of singlet Oxygen when exposed to solar radiation. In this situation, the use of anti-singlet Oxygen materials in sunscreen formulations becomes a necessity.

## Figures and Tables

**Table 1 antioxidants-09-01091-t001:** Reactive oxygen species relevant to skin damage.

ROS	Biochemical/Biophysical Origin
Hydrogen peroxide	Photosensitization of aromatic amino acids (Tyr, Trp)
Hydrogen peroxide	Oxidative burst of inflammatory cells
Hydroxyl radical	Fenton reaction
Superoxide	Charge transfer to molecular oxygen upon absorption of a UV photon by a photosensitizer
Singlet Oxygen	Energy transfer to molecular oxygen upon absorption of a UV photon by an endogenous photosensitizer
Singlet Oxygen	Energy transfer to molecular oxygen upon absorption of a UV photon by **commercial sunscreens**

**Table 2 antioxidants-09-01091-t002:** Reactive oxygen species and appropriate antioxidants.

Reactive Oxygen Species	Counteracting Molecules
Hydroxyl radical OH	Mannitol
	Ergothioneine
Peroxy radical LOO	α-tocopherol
superoxide ion O_2_^−^	α-tocopherol
Peroxynitrite ONOO-	Bakuchiol
singlet Oxygen ^1^O_2_	β-Carotene
	Xanthine, Astaxanthine, Lycopene etc.
	Acetyl Zingerone
	Bakuchiol
	Micah

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
