# Peer review of "Appropriate Technologies to Accompany Sunscreens in the Battle Against Ultraviolet, Superoxide, and Singlet Oxygen"

_antioxidants, 2020, doi:10.3390/antiox9111091_

Round 1

Reviewer 1 Report

To authors

Comment 1. The description of antioxidant is insufficient in the introduction. As many antioxidant studies have been conducted so far, related references and descriptions of natural antioxidants are needed.

Comment 2. Describe the side effects of synthetic antioxidants and the need for natural antioxidants in the introductory section. This will allow readers to recognize the importance of natural antioxidants over synthetic products.

Comment 3. Table 2 describes ROS and antioxidants suitable for each ROS. However, it is judged that additional entry of natural antioxidants involved in various ROS is necessary by citing several other references.

Comment 4. It is thought that it would be better if the various organisms and cases developed as products involved in the sunscreen are described in the contents. A well-organized table with regard will enhance the readability of the publication.

Author Response

Anwer to the comments To authors by Reviewer n 1. (in Italic, below)

Comment 1. The description of antioxidant is insufficient in the introduction. As many antioxidant studies have been conducted so far, related references and descriptions of natural antioxidants are needed.

The Introduction of the manuscript does not contain any definition of anti-oxidant (that by itself might need a longer review) and does not contain any reference to natural anti-oxidants.

Comment 2. Describe the side effects of synthetic antioxidants and the need for natural antioxidants in the introductory section. This will allow readers to recognize the importance of natural antioxidants over synthetic products.

Anti-oxidants, as everything else, can be toxic when used at concentrations above a certain threshold (Paracelsus pointed out that everything is toxic, it only depends on the dose) and we know that too much antioxidant becomes pro-oxidant. So, the side effects of anti-oxidants do not depend on their origin but on their concentration. The goal of this manuscript is not to praise natural anti-oxidants versus synthetic ones: the goal of this manuscript is to describe technologies appropriate to fight Superoxide and singlet Oxygen

Comment 3. Table 2 describes ROS and antioxidants suitable for each ROS. However, it is judged that additional entry of natural antioxidants involved in various ROS is necessary by citing several other references.

Table 2 gives an example of molecules that are particularly "active" in targeting one ROS more than others, The table is just an example. "Natural" antioxidants may or may not have such a pronounced "preference" for this more that for that ROS. The manuscript is not a review of antioxidants in general. The manuscript describes a few technologies to hinder Superoxide and singlet Oxygen. And by the way, strictly speaking, a molecule that hinders the formation of singlet Oxygen could perhaps be considered as non belonging to the anti-oxidant category.

Comment 4. It is thought that it would be better if the various organisms and cases developed as products involved in the sunscreen are described in the contents. A well-organized table with regard will enhance the readability of the publication.

The manuscript does not describe the formulation of sunscreens. It reports that some UV filters such as ZnO or TiO2 or one of the Mexoryls, do generate singlet Oxygen when exposed to UV radiation. A table with ingredients used to formulate sunscreens would necessarily be incomplete and would not add to the understanding of the main message of the manuscript.

Sincerely

Paolo Giacomoni

Reviewer 2 Report

This manuscript carefully describes the biochemical consequences of the interactions between UV light and sunscreens and the following reactions triggered by the reaction products derived from UV/sunscreen interaction. Superoxide anions and singlet oxygen are in the focus of considerations. It is a great intellectual pleasure to read this manuscript.

I have a few suggestion for further improvement of this manuscript:

1) page 2, lines 73, 74 .."release of singlet oxygen produced by MPO...": This sentence may be misleading, as it might be understood as enzymatic generation of singlet oxygen by MPO directly. To my knowledge, singlet oxygen is generated in this setting through the interaction between HOCl (one of the reaction products of MPO) and H2O2. Please clarify.

2) p3, lines 100, 101: ..."superoxide scavengers like alpha-tocopherol...". Please mention the type of reaction between superoxide anions and alpha-tocopherol and mention that the interaction between alpha-tocopherol with lipid peroxides is its dominant function.

3) p5, line 207: "reaction of peroxynitrite with the cytoskeleton". This is correct in principle. However, peroxynitrite potentially generated in UV-treated sunscreen on the skin has no chance to reach the interior of the cell. The major part of peroxynitrite generated at this site will react with CO2, leading to the formation of nitrosoperoxycarboxylate. Another part may reach the surface of the cell layer. It proton pumps are present, peroxynitrite will be protonated to peroxynitrious acid and subsequently decompose into NO2 and potentially damaging hydroxyl radicals (please see Bauer, Redox Biol 6: 353-371, 2015.)

4) p6, line 255: "among the best known ROS, singlet oxygen is perhaps the most reactive one...". I assume that hydroxyl radicals are the most reactive ROS. However, singlet oxygen (besides its moderated potential for induction of lipid peroxidation) has a great potential to control redox signaling pathways. This is due to its potential to inactivate catalase and SOD through reaction with histidine in the active centers of these enzymes (Escobar et al., Free Radic Biol Med, vol. 20, pp. 285-290, 1996.; Kim et al. Biochimie, vol. 83, pp. 437-444, 2001). One of the consequences of this interaction are specific cell death inducing effects on malignant cells (Riethmüller et al., Redox Biol, 6 (2015) 157-168.). Therefore, singlet oxygen generated in UV-illuminated sunscreen might act analogous to photodynamic antitumor therapy (certainly on a low and so far merely hypothetical level). But this would be a positive effect of singlet oxygen.

Another consequence based on singlet oxygen generation in UV-illuminated sunscreen might be a damaging effect on bacteria. Singlet oxygen has been shown to inactivate bacteria very efficiently (Dahl TA et al.,  Photochemistry and Photobiology 1987, 46: 345-352; Maisch Tet al.,  Proc Natl Acad Sci USA 2007; 104: 7223-7228; Tatsuzawa H, et al., FEBS Letters 1998, 439: 329-333). This effect might inactivate unwanted bacteria and prevent local infection, but more likely, I assume, it might have a damaging effect on the normal bacterial flora of the skin. As the normal flora has an important function for protection of the skin, the removal of singlet oxygen would be advisable.

Author Response

Answer to the comments of reviewer n 2,

I would like to thank the reviewer for the nice words and for the relevant comments. As a consequence of the comments I have introduced the following changes into the manuscript.

1-Lines 75-76 the sentence has been changed to: There is also a release of Singlet Oxygen when the immune cells cross the dermis (23)

2-Lines 103 and following: the sentence has been changed to:

...such as alpha tocopherol that is oxidized to produce alpha-tocopheryl quinone, alpha tocopherol dimer and alpha-tocopherol dihydroxy dimer,

3-Point taken. We speak here of the Peroxinitrite that is generated within the cells by the reaction occurring between endogenous NO and endogenous photo-sensitizer-produced Superoxide. Lines 212 and following the sentence has been modified to: Another ROS, peroxinitrite, is an oxidant produced by the reaction between the endogenously produced  NO and the photo-sensitizer-produced Superoxide radicals (O2*-) that is effective in inducing lipid peroxidation, damaging DNA and oxidizing sulfhydryl groups in proteins.

4-I agree. When singlet Oxygen is generated in situ within tumor cells fed with ALA, as it is the case in PDT, it kills the tumor cells and can have a therapeutic value. When singlet Oxygen is randomly produced in or near healthy cells, it can kill them and trigger an inflammatory reaction as well as the synthesis of matrix metallo-proteinases that digest the elastic fibers of the dermis and accelerate the process of skin aging.

And yes, skin micro-flora can also be damaged by singlet Oxygen generated by sunscreens: I do not have first hand data to quote, I can only say that I worked on the anti-microbialeffect of singlet Oxygen, see for instance:

Merchat, Giacomoni, Villanueva, Bertoloni & Jori (1995) Photo-sensitization of Bacteria to Visible Light by Meso-substituted Porphyrins, J. Braz. Chem. Sci. 6 : 123-125 but I did not think it was relevant to quote such results in this manuscript. Anyway, the sentence line 261 has been modified to: As all the known Reactive oxygen Species, singlet Oxygen is a very reactive moiety

Round 2

Reviewer 1 Report

It is considered appropriate to publish the article in the Antioxidants journal.